# Rootstock Effects on Fruit Yield and Quality of ‘BRS Tainá’ Seedless Table Grape in Semi-Arid Tropical Conditions

**DOI:** 10.3390/plants13162314

**Published:** 2024-08-20

**Authors:** Carlos Roberto Silva de Oliveira, Antônio Francisco de Mendonca Junior, Patrícia Coelho de Souza Leão

**Affiliations:** 1Department of Agronomy, Universidade Federal Rural de Pernambuco, Recife 52171-900, PE, Brazil; carlos-robertoliveira@hotmail.com (C.R.S.d.O.); antonio.mendoncajunior@ufrpe.br (A.F.d.M.J.); 2Embrapa Semiárido, Petrolina 56302-970, PE, Brazil

**Keywords:** *Vitis vinifera*, tropical viticulture, seedless table grapes

## Abstract

In viticulture, choosing the most suitable rootstock for a specific scion cultivar is an efficient and cost-effective way to increase yield and enhance the physicochemical characteristics of the fruit. The objective of this study was to evaluate the agronomic performance of the ‘BRS Tainá’ grapevine on different rootstocks under the conditions of the Sub-Middle São Francisco Valley. The main experimental factor consisted of eight rootstocks (IAC 313, IAC 572, IAC 766, 101-14 MgT, Paulsen 1103, Ramsey, SO4, and Teleki 5C), arranged in randomized blocks with four replicates. The experiment was conducted over four production cycles, from 2021 to 2023, in a commercial crop area in Petrolina, PE, Brazil. There were significant effects of rootstocks for the yield and number of bunches per plant, as well as berry length and firmness. ‘BRS Tainá’ achieved the highest yield (22.2 kg per plant) when grafted onto the Paulsen 1103 rootstock, which was superior to the yield on 101-14 MgT, IAC 313, and IAC 572 rootstocks. The highest number of bunches (88) was obtained with ‘BRS Tainá’ grafted on Paulsen 1103, while the lowest number (63) was obtained on IAC 572; both these rootstocks were not significantly different from the other rootstocks. For all scion–rootstock combinations, the mean values for soluble solid (SS) content, titratable acidity (TA), and the SS/TA ratio were similar to those previously described for ‘BRS Tainá’, meeting the commercialization standard. The results for the yield and number of bunches per plant indicate the suitability of grafting ‘BRS Tainá’ on Paulsen 1103 under the semi-arid tropical conditions of the São Francisco Valley.

## 1. Introduction

The São Francisco Valley, located in the Brazilian Northeast region, is the main producer and exporter of table grapes in Brazil [1]. In 2022, a volume of about 396.7 thousand tons was harvested in this region, generating BRL 2 billion in production value, almost two-thirds of the total table grape value of Brazil [2]. Irrigated fruit growing in this region is favored by the semi-arid tropical climate; the availability of water resources; investments in research; and the organization of the production chain, supply of inputs, and qualified professionals, which makes it possible to obtain two harvests per year.

In viticulture, the most suitable rootstocks should be chosen when planning the establishment of the vineyard, as the affinity between scion and rootstock tends to affect the vigor, yield components, and quality of the grapes produced [3,4,5]. In addition, there is a strong interaction between the rootstock and the edaphic and climatic conditions of each region and particular property, with implications for vineyard management. In Brazil, table grapes grown under semi-arid tropical conditions are commonly grafted onto IAC 313, IAC 572, and IAC 766 Brazilian rootstocks, which are perfectly adapted to tropical climatic conditions and different types of soil [6]. In addition, Teleki 5C, 101-14 MgT, Ramsey (Salt Creek), Paulsen 1103, and SO4 rootstocks, which provide medium-to-high tolerance to root-knot nematodes, have been used in commercial vineyards [6,7].

The use of different rootstocks has led to changes in the morphoagronomic and physicochemical variables of different grape cultivars grown in the São Francisco Valley region, such as Brasil, Benitaka, Itália, BRS Vitória, Thompson Seedless [8,9,10], and Chenin Blanc [11], as well as the juice grape cultivars BRS Cora, BRS Magna, and Isabel Precoce [12,13,14]. However, the high market demand for new seedless table grape cultivars has not allowed for studies on the identification of the ideal rootstock or group of rootstocks for each scion cultivar before new cultivars are released.

‘BRS Tainá’ is an early maturing variety and a white-colored seedless grape with medium-sized berries, firm flesh texture, and a pleasant neutral flavor [15]. It is the first grape cultivar developed by the Embrapa program ‘Uvas do Brasil’ under semi-arid tropical environmental conditions [15]. This cultivar was created to make up for the lack of a high-yielding white table grape varieties adapted to the environmental conditions of a semi-arid region that could be grown by small and medium-sized producers without restrictions on the expansion of growing areas and without the need to pay royalties imposed by private international companies of genetic material.

In a previous study, no significant changes were observed in the vigor and fertility of buds from ‘BRS Tainá’ when grafted on different rootstocks [4], but there is still no information related to fruit yield and quality. Thus, the objective of this study was to determine the effect of different rootstocks on the yield components and physicochemical characteristics of ‘BRS Tainá’ grapes grown under irrigated conditions in a Brazilian semi-arid tropical region.

## 2. Materials and Methods

### 2.1. Experimental Area and Growing Conditions

The experiment was carried out in a commercial vineyard located in Petrolina, PE, Brazil (9°19′ S, 40°28′ W, and 386 m altitude) over four production cycles in the 2021–2023 period. Production pruning and harvest dates of each production cycle are presented in Table 1. The soil is classified as an Argissolo Amarelo (Ultisol). The climate of the region is classified as BSh’ (very hot and dry) according to Koppen’s classification [16]. Monthly climate data throughout the experimental period are presented in Figure 1. One-year-old ‘BRS Tainá’ grapevines (first production pruning) were grown in a Pergola horizontal vine training system, with a plant spacing of 3.5 × 2.5 m; two plants were planted side by side, and it had a single-cordon canopy architecture. A mixed-type production pruning was adopted, with canes (five to six buds) and spurs (two or three buds), which corresponded to a mean bud load value of 22 buds m^−2^.

Drip irrigation was performed based on reference evapotranspiration information. Regardless of rainfall, the total amount of irrigation water in the four production cycles was 384 mm, 436 mm, 380 mm, and 468 mm, respectively. The physicochemical properties of the soil are shown in Table 2. Fertilization was performed through fertigation according to the nutritional needs observed in soil analysis [17]. In all production cycles, measures of 60 kg N ha^−1^, 60 kg P ha^−1^, and 160 kg K ha^−1^ were applied as fertilizers. Crop practices related to scion management included shoot thinning, defoliation, tying, shoot topping, and bunch and berry thinning. In the moderately compact bunches, thinning was carried out on around 20% of the berries when they reached pea size (7 mm). The soil was mechanically tilled or hoed for weed control, while the main pests and diseases in the region were controlled through chemical spraying to prevent severe infestations and heavy losses.

### 2.2. Experimental Design

‘BRS Tainá’ grapevines were grafted onto IAC 313, IAC 572, IAC 766, 101-14 MgT, Paulsen 1103, Ramsey (Salt Creek), SO4, and Teleki 5C rootstocks. The experimental design was randomized blocks, with four replicates (plots) for each scion–rootstock combination (cv. ‘BRS Tainá’ × eight rootstocks), in a split-plot arrangement, with the production cycles constituting the plots and the main factor (rootstocks) constituting the split plots. Each plot included four grapevines, and one of the two central plants in each plot was evaluated. Rootstocks were selected based on the changes observed in morphoagronomic and physicochemical variables in different grape cultivars grown in the São Francisco Valley region, as well as based on rootstock availability and grower preference.

### 2.3. Analyzed Variables 

The following variables were evaluated in all the production cycles: yield per plant (YP) (in kg), determined by collecting and weighing all bunches of the plant after thinning; the number of bunches per plant (NB), obtained during harvest by counting; bunch weight (BWt) (in g), bunch length (BL) (in cm), and bunch width (BW) (in cm), determined using five bunches per plot; and berry mass (BeW) (in g), berry length (BeL) (in mm), and berry diameter (BeD) (in mm), determined with a scale and a graduated ruler, using 10 berries from each bunch, for a total of 50 berries per plot. Physicochemical variables and firmness were measured during the first three production cycles as follows: The soluble solid (SS) content, in ºBrix, was determined in the grape extracted from 50 berries using a digital refractometer with automatic temperature adjustment (ATAGO, Digital Pocket Refractometer, model PAL-1); titratable acidity (TA), expressed in g of tartaric acid.100 mL^−1^, was determined using 5 mL of grape pulp in 50 mL of distilled water, with a 0.1 M NaOH solution for titration [18]; the SS/TA ratio; and berry firmness (F), expressed in Newton (N), was obtained from 15 berries taken from the apical, median, and basal parts of 5 bunches per plot, using an Extralab TA.XT.Plus digital texture analyzer (Stable Micro Systems, Surrey, London, UK).

### 2.4. Statistical Analysis

Analysis of variance was carried out on the data using the Shapiro–Wilk test. When the analysis indicated statistical significance, the values of the means of the main factor (rootstock), production cycles, and the interaction between these factors were compared using Tukey’s test at the 5% probability level. The yield, bunch weight, and number of bunches variables were transformed using $\sqrt{x}$. These analyses were performed using the SISVAR 5.6 program [19].

## 3. Results and Discussion

There was no significant effect of the interaction between rootstock and production cycles for the yield per plant, the number of bunches per plant, and bunch weight variables. For yield per plant, the average performance of ‘BRS Tainá’ grapevines ranged from 11.88 to 22.2 kg per plant, and the values obtained for grapevines grafted on the Paulsen 1103 rootstock were higher than those obtained with 101-14 Mgt, IAC 313, and IAC 572 (Table 3), achieving an average yield of 25 tons/ha on this rootstock with two harvests per year.

Regarding the number of bunches, the results from most rootstocks had similar values, and a significant difference was observed only between Paulsen 1103 and IAC 572, with means of 88 and 63 bunches per plant before thinning, respectively. In this study, the yield performance of ‘BRS Tainá’ was satisfactory when it was grafted on Paulsen 1103, Ramsey, SO4, Teleki 5C, and IAC 766. These five rootstocks share common characteristics, with medium-to-high levels of traits such as the vegetative vigor of the scion cultivar, resistance to nematodes, and tolerance to salinity [8,20,21,22]. In studies with the white table grapes ‘Sugraone’ and ‘BRS Clara’, produced in the São Francisco Valley, the rootstocks Paulsen 1103 and SO4 also showed the highest means for both yield and number of bunches per plant [23,24], and the means of ‘BRS Tainá’ were higher than the means of these two cultivars for both rootstocks. In the Coquimbo Region of Chile, the Ramsey (Salt Creek) rootstock led to an increase in the yield and number of bunches in the Thompson Seedless cultivar compared to the Paulsen 1103 rootstock [25]. Thus, it is evident that the yield performance of grafted grapevines depends on several factors, such as the compatibility of the scion and the rootstock, management practices, and the edaphic and climatic characteristics of the crop site.

The mean values of yield, the number of bunches per plant, and bunch weight with all the rootstocks (Table 3), except for yield on IAC 572, were similar to or higher than the values reported by Leão et al. (2021) [15], who stated that, under favorable growing conditions, the expected means of the ‘BRS Tainá’ cultivar are 14.5 kg for yield per plant, about 55 for the number of bunches per plant, and around 300 g for bunch weight. The differences in YP among the rootstocks may be related to the NB obtained and the intrinsic characteristics of the rootstock since the lowest YP values were observed in the 101-14 MgT, IAC 313, and IAC 572, which tend to promote very high canopy vigor and, consequently, a decrease in yield [6,25]. Regardless of the rootstock used, the overall mean of YP for ‘BRS Tainá’ was similar to that of the table grape cultivars Flame Seedless (±15.0 kg), Thompsom Seedless (±15.5 kg), BRS Vitória (16.6 kg), Arizul (15.4 kg), and BRS Clara (16.1 kg) [8,25,26].

The season factor significantly affected yield. In the 2022.2 production cycle, the grapevines exhibited a higher yield than in 2021.2, indicating that the greater age of the vines had a greater effect on yield than the rootstocks used for grafting. As of the third year of production (2023.1 and 2023.2), ‘BRS Tainá’ grapevines achieved the highest means for yield per plant and the number of bunches (Table 3). The 2023.2 production cycle was the only one in which there was no rainfall between production pruning and harvest (Table 1 and Figure 1). The higher temperatures and solar radiation may have favored increased photosynthetic rates and carbohydrate accumulation, resulting in a higher yield [27]. During the 2023.1 production cycle, the losses brought about by rainfall between the berry growth and fruit ripening stages and the high number of bunches per plant (129.0) may have been the main factors that contributed to the reductions in yield and bunch weight [28]. Although bunch weight had a higher mean in the 2022.2 production cycle, the yield and number of bunches per plant showed lower values compared to 2023.2. The number of bunches in 2022.2 was about half the number produced in 2023.2, which may have favored a better distribution of carbohydrates to the bunches, due to less competition among them. According to Leão and Oliveira (2023) [24], who studied five table grape cultivars in the São Francisco Valley, there is usually an inverse correlation between the number of bunches per plant and bunch weight; the tendency is that the greater the number of bunches per plant, the lower the weight of these bunches. This result points to the importance of thinning and selecting bunches to obtain better quality fruit; reducing the number of bunches in accordance with plant vigor is recommended.

The production cycles had significant effects on bunch length, bunch width, berry weight, and berry diameter (Figure 2). According to Leão et al. (2021) [15], the bunches of ‘BRS Tainá’ have an average size of about 17.0 cm length and 11.0 cm width, along with medium-sized berries (±5.0 g), with length and diameter around 24.0 × 19.0 mm. Higher values may be achieved when growth regulators are applied for bunch elongation and berry growth. All the physical bunch and berry variables showed significant differences for years and growing seasons, confirming that the climatic variations among the production cycles were more responsible for these changes than the use of different rootstocks. The 2021.2, 2022.2, and 2023.2 cycles had higher temperatures, which favored an increase in photosynthetic rates and the accumulation of carbohydrates, resulting in an increase in berry and bunch weights and sizes. Furthermore, when the rainy season occurs between the berry growth stage and fruit harvest, a reduction in these variables is even more pronounced, as seen in the 2023.1 production cycle.

Berry length was the only morphological variable significantly affected by the interaction between rootstocks and production cycles (Table 4). The fruit showed greater stability for berry length among production cycles when ‘BRS Tainá’ was grafted onto Paulsen 1103, Ramsey, and SO4. Only in the 2023.1 cycle were there significant differences in berry length among rootstocks, with Paulsen 1103 and SO4 higher than IAC 572; the same result was obtained for the mean of the production cycles. Regarding the production cycles, the berries were larger in 2021.2 and 2022.2, when the lowest values for the number of bunches per plant were obtained (Table 3), which reduced the competition among bunches, and consequently among berries, for photoassimilates, favoring greater growth [8].

Regarding berry firmness (F), ‘BRS Tainá’ showed firmer berries when grafted on IAC 572 (6.88 N) than when grafted on 101-14 MgT (6.27 N), and neither rootstock differed from the other rootstocks (Figure 3). There were significant differences among the production cycles, with the lowest value obtained during the first production cycle (5.21 N) and the highest obtained in 2023.1 (7.85 N), while 2022.2 had an intermediate value (6.58 N). It is important to highlight that the means observed for production cycles, as well as the overall mean of each rootstock, were considered high and desirable since firmer berries tend to reduce losses caused by mechanical damage during fruit storage and transport. All the values observed for ‘BRS Tainá’ grafted onto rootstocks and during the production cycles were similar to those previously reported, as well as for the cultivars ‘Thompson Seedless’ (5.58 N), ‘Brasil’ (5.0 N ± 1.8), ‘Benitaka’ (5.5 ± 1.8 N), and ‘Itália’ (5.7 ± 1.7 N), grown in the same region [9,15].

The effect of the rootstock–production cycle interaction was significant for all physicochemical variables (Table 5). For the soluble solid content, ‘BRS Tainá’ grapes showed higher mean values in the 2023.1 production cycle, regardless of the rootstock used. The mean values ranged from 15.29 ºBrix (2022.2) to 19.85 ºBrix (2023.1), with the highest value achieved in the production cycle in which the berries had the lowest weight and size (Figure 2 and Table 4). Despite the differences in the means of the rootstock effects among the production cycles, the overall mean considering the three production cycles did not show significant differences. Thus, according to Brazilian legislation [29], all the values of the soluble solid content observed in ‘BRS Tainá’ grapes were higher than the minimum required (14 ºBrix) for Brazilian and international commercialization.

There were significant differences in titratable acidity only among the production cycles, with the highest mean values achieved in the 2022.2 cycle (Table 5). The rain that occurred during the berry ripening and harvest period, in October 2022 (Figure 1), may have reduced temperatures in the berries and, consequently, affected the metabolic processes that degrade acids, resulting in fruit with higher acidity [27]. Although ‘BRS Tainá’ berries have a neutral flavor and intermediate acidity (0.47–0.54), during the 2021.2 and 2023.1 cycles, the TA mean values were lower than expected for this cultivar in the São Francisco Valley [15]. In addition, the mean values of all cycles were similar to those observed for other table grapes, such as ‘Arizul’, ‘BRS Clara’, ‘BRS Linda’, ‘BRS Melodia’, ‘Marroo Seedless’, and ‘Thompson Seedless’, grown under the semi-arid tropical conditions of the São Francisco Valley [24,26].

The SS/TA ratio ranged from 28.15 (2022.2) to 60.44 (2023.1). According to Benato (2003) [30] and Lima and Choudhury (2007) [31], mean values ≥ 20.0 are necessary to prolong the storage and shelf life of seedless table grapes, but values higher than 45.0 are not desired, as they can interfere in the flavor and reduce shelf life [32]. Thus, despite the superiority in the SS/TA ratio in the 2023.1 production cycle, the mean values affected by rootstocks were considered above the desirable level. However, in the overall mean of production cycles, there was no difference among rootstocks, so the mean value of 44.32 can be considered appropriate for the commercialization and conservation of the physicochemical quality of the grapes.

## 4. Conclusions

The production cycles led to significant changes in all 12 morphological and physicochemical variables studied, with cycles in the second half of the year resulting in larger bunches and berries but with a lower soluble solid content. In the present study, bunch weight, length, and width, as well as berry weight and diameter, were not affected by rootstocks. The physicochemical characteristics of soluble solid (SS) content, titratable acidity (TA), and the SS/TA ratio showed no significant changes when vines were grafted on different rootstocks. Grafting ‘BRS Tainá’ onto Paulsen 1103, Ramsey, SO4, and Teleki 5C is recommended to increase vineyard yield in soil and tropical climatic conditions similar to those in this study. The lowest yield was recorded in grapevines grafted on the IAC 313, IAC 572, and 101-14 MgT rootstocks.

## Figures and Tables

**Figure 1 plants-13-02314-f001:**
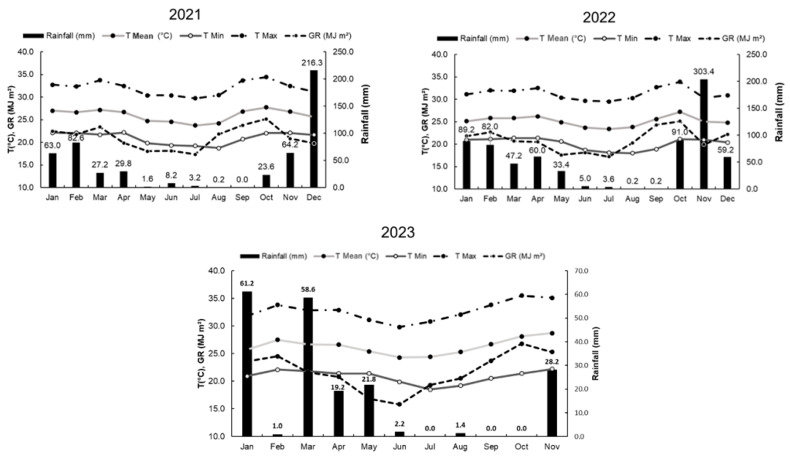
Monthly averages of minimum, mean, and maximum air temperature (T, in °C), rainfall (mm), and global radiation (GR, in MJ/m^2^) from 2021 to 2023, in Petrolina, PE, Brazil.

**Figure 2 plants-13-02314-f002:**
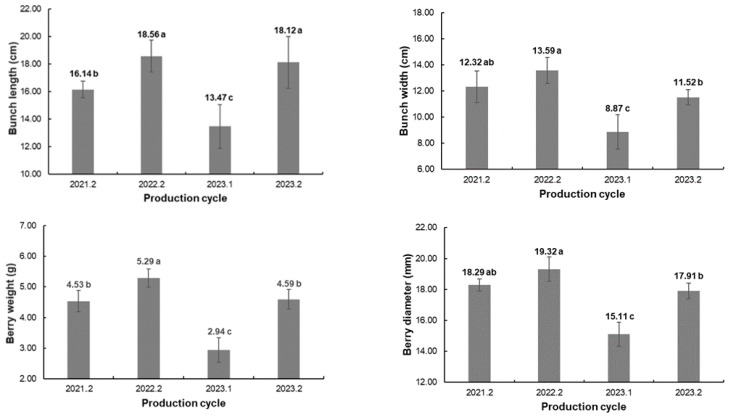
Means and standard deviations of bunch length, bunch width, berry weight, and berry diameter of ‘BRS Tainá’ grapevines over four production cycles in Petrolina, PE, Brazil. Mean values followed by the same lowercase letters in the bar comparing production cycles do not differ from each other by Tukey’s test (*p* < 0.05).

**Figure 3 plants-13-02314-f003:**
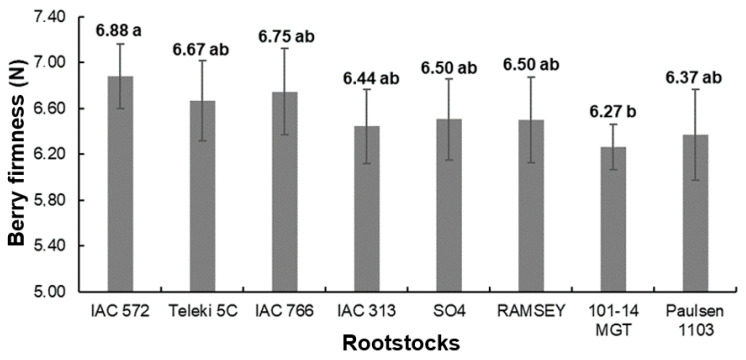
Means and standard deviations of berry firmness (F) of ‘BRS Tainá’ grapevines over three production cycles in Petrolina, PE, Brazil. Mean values followed by the same lowercase letters in the bar comparing production cycles do not differ from each other by Tukey’s test (*p* < 0.05).

**Table 1 plants-13-02314-t001:** Production pruning and harvest dates during the four production cycles evaluated.

Production Cycle	Production Pruning	Harvest
2021.2	23 August 2021	7 December 2021
2022.2	7 July 2022	31 October 2022
2023.1	12 January 2023	26 April 2023
2023.2	6 July 2023	26 October 2023

**Table 2 plants-13-02314-t002:** Physicochemical properties of the soil.

Soil Properties	Soil Layers Value	
	(0–20 cm)	(20–40 cm)
Sand content (g kg^−1^)	784.88	778.85
Silt content (g kg^−1^)	189.24	149.07
Clay content (g kg^−1^)	55.88	70.18
Soil bulk density (kg dm^−3^)	1.31	1.34
Porosity (%)	48.37	47.71
Organic matter (g kg^−1^)	7.0	3.9
ECE (mS cm^−1^)	0.71	0.56
pH water	5.5	5.4
Ca (cmol_+_dm^−3^)	2.4	2.3
Mg (cmol_+_ dm^−3^)	2.8	2.7
Na (cmol_+_ dm^−3^)	0.03	0.02
Al (cmol_+_ dm^−3^)	0.00	0.00
K (cmol_+_ dm^−3^)	0.25	0.17
P (mg dm^−3^)	120.11	130.26

ECE—electrical conductivity of soil saturation extract; pH water—1:2.5 soil–water ratio; Ca—calcium; Mg—magnesium; Na—sodium; Al—aluminum; K—potassium; P—phosphorus.

**Table 3 plants-13-02314-t003:** Means and coefficients of variation for yield per plant (YP); number of bunches per plant (NB); and bunch weight (BWt) of ‘BRS Tainá’ grapevines over four production cycles in Petrolina, PE, Brazil.

Rootstock	YP (kg)	NB	BWt (g)
101-14 MgT	15.86 b ^1^	81.69 ab	335.21 ^ns^ *
IAC 313	14.55 b	86.56 ab	321.96
IAC 572	11.88 b	62.75 b	323.01
IAC 766	16.46 ab	84.53 ab	335.97
Paulsen 1103	22.20 a	87.75 a	350.28
Ramsey	16.80 ab	79.88 ab	324.57
SO4	16.65 ab	79.56 ab	340.07
Teleki 5C	16.78 ab	80.53 ab	341.00
Mean	16.40	80.41	334.01
CV (%)	18.31	13.03	9.87
Production cycle			
2021.2	10.79 c	37.94 c	318.13 b
2022.2	15.09 b	49.44 c	479.15 a
2023.1	15.94 b	129.25 a	177.85 c
2023.2	23.78 a	105.00 b	360.92 b
CV (%)	16.08	13.64	11.22

^1^ Means followed by the same letter in the column do not differ from each other by Tukey’s test at the 5% probability level. ns * = not significant.

**Table 4 plants-13-02314-t004:** Means of berry length as affected by the interaction between rootstocks and production cycles of ‘BRS Tainá’ grapevines in Petrolina, PE, Brazil.

Rootstock	Berry Length (mm)				
	2021.2	2022.2	2023.1	2023.2	Mean
101-14 MgT	22.88 aAB ^1^	24.70 aA	21.93 abB	21.62 aB	22.78 ab
IAC 313	23.11 aAB	24.35 aA	21.04 abBC	20.64 aC	22.28 ab
IAC 572	23.23 aAB	24.22 aA	19.80 bC	21.24 aBC	22.12 b
IAC 766	23.25 aAB	23.92 aA	21.11 abC	21.17 aBC	22.36 ab
Paulsen 1103	23.06 aA	24.32 aA	22.35 aA	22.49 aA	23.05 a
Ramsey	23.16 aA	24.04 aA	22.75 aA	22.26 aA	23.05 a
SO4	22.94 aA	23.41 aA	21.48 abA	22.67 aA	22.62 ab
Teleki 5C	23.45 aA	24.27 aA	20.93 abB	22.51 aAB	22.79 ab
Mean	23.13 A	24.15 A	21.42 B	21.82 B	22.63
CV					3.96

^1^ Means followed by the same lowercase letters in the column, comparing rootstocks, and the same uppercase letters in the row, comparing production cycles, do not differ by Tukey’s test (*p* < 0.05). CV = coefficient of variation.

**Table 5 plants-13-02314-t005:** Means of the soluble solid (SS) content, titratable acidity (TA), and the SS/TA ratio as affected by the interaction between rootstocks and production cycles of ‘BRS Tainá’ grapevines over three production cycles in Petrolina, PE, Brazil.

**Rootstock**	**Soluble Solids (ºBrix)**			
	**2021.2**	**2022.2**	**2023.1**	**Mean**
101-14 MgT	16.00 aB ^1^	14.75 aB	20.55 aA	17.10 ^ns^ *
IAC 313	14.53 aB	15.60 aB	20.25 aA	16.79
IAC 572	15.73 aB	16.03 aB	19.15 aA	16.97
IAC 766	15.05 aB	15.30 aB	20.00 aA	16.78
Paulsen 1103	16.38 aB	15.28 aB	19.45 aA	17.03
Ramsey	15.18 aB	15.20 aB	19.63 aA	16.67
SO4	15.05 aB	14.98 aB	20.38 aA	16.80
Teleki 5C	15.40 aB	15.20 aB	19.38 aA	16.66
Mean	15.41 B	15.29 B	19.85 A	16.85
CV (%)				5.97
**Rootstock**	**Titratable acidity (g of tartaric acid 100 mL^−1^)**			
	**2021.2**	**2022.2**	**2023.1**	**Mean**
101-14 MgT	0.36 aB	0.56 aA	0.33 aB	0.42 ^ns^
IAC 313	0.33 aB	0.51 aA	0.36 aB	0.40
IAC 572	0.36 aB	0.57 aA	0.37 aB	0.43
IAC 766	0.37 aB	0.56 aA	0.37 aB	0.43
Paulsen 1103	0.36 aB	0.57 aA	0.30 aB	0.41
Ramsey	0.31 aB	0.53 aA	0.40 aB	0.41
SO4	0.36 aB	0.54 aA	0.30 aB	0.40
Teleki 5C	0.39 aB	0.56 aA	0.28 aB	0.41
Mean	0.35 B	0.55 A	0.34 B	0.41
CV (%)				9.88
**Rootstock**	**SS/TA ratio**			
	**2021.2**	**2022.2**	**2023.1**	**Mean**
101-14 MgT	44.70 aB	26.44 aC	62.44 abcA	44.52 ^ns^
IAC 313	45.17 aB	31.10 aC	59.08 abcA	45.12
IAC 572	44.24 aB	28.15 aC	53.13 bcA	41.84
IAC 766	41.09 aB	27.67 aC	54.13 bcA	40.96
Paulsen 1103	46.23 aB	26.97 aC	65.58 abA	46.26
Ramsey	50.91 aA	28.92 aB	50.45 cA	43.42
SO4	42.87 aB	28.50 aC	69.63 aA	47.00
Teleki 5C	39.85 aB	27.51 aC	69.10 aA	45.86
Mean	44.38 B	28.15 C	60.44 A	44.32
CV (%)				11.97

^1^ Means followed by the same lowercase letters in the column, comparing rootstocks, and the same uppercase letters in the row, comparing production cycles, do not differ by Tukey’s test (*p* < 0.05). ^ns^ * = not significant. CV = coefficient of variation.

## Data Availability

The datasets generated and/or analyzed during the current study are available from the corresponding author upon reasonable request.

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
