# Peer review of "Rootstock Effects on Fruit Yield and Quality of ‘BRS Tainá’ Seedless Table Grape in Semi-Arid Tropical Conditions"

_plants, 2024, doi:10.3390/plants13162314_

Round 1

Reviewer 1 Report (New Reviewer)

Comments and Suggestions for Authors

This is an interesting paper assessing the influence of several rootstocks on Fruit Yield and Quality of ‘BRS Tainá’ 2 Seedless Table Grape in Tropical Semi Arid Conditions. Although this manuscript is well-written and uses an appropriate experimental plan and statistical analysis, the other hand, Authors are missing some important information on plant material and data discussions. The references can also be improved.

L 68. Material and methods – 2.1 Experimental area and cultivation conditions

L. 72Authors give a soil classification and report in L-80-81: “Fertilization was performed through fertigation according to the nutritional needs observed in soil analysis” To better understand I suggest reporting soil analysis and table with the Units of fertilizer applied. The same consideration for water management, at least please add the total amount of applied water.

L102 : please report when bunch thinning was done and the criteria.

L 124 Results and Discussion

Can Authors report the percentage of bunch thinning?

In fact,  according to the reported literature (L156-157) Authors should find 14.5 kg of yield /vine with 300 g of bunch mass. How can this explain differences among all rootstock combinations?  However in Table 2, during 2023.1 production cycle, the Authors recorded more bunches/vine with less weight. These bunches were more compact in fact they had more berries, and less bunch length. A bunch mass equal to 177.85 g, compared to the other bunches in the different production cycles, are they different in compactness, please add a comment in the discussion section.

L161-163 ……. from the 2022.2 the grapevines exhibited higher production and higher bunch mass, in relation to 2021.2 ……

L180-182: Authors report: “This result points to the importance of thinning and selecting bunches to obtain better quality fruits, so it is indicated to reduce the number of bunches according to the vigor of the plant.” For this reason, it is important to report the thinning criteria.

L191-193: Authors report: “The 2021.2, 2022.2, and 2023.2 cycles had higher temperatures, which favored an increase in photosynthetic rates and the accumulation of carbohydrates, resulting in an increase in berry and bunch weights and sizes”. This is true, moreover, in 2023.1 there were more berries/bunch

Why did the Authors at a certain point miss information about 2023.2 production cycle? (e.a. table 4 and in the discussion)?

L 269 Conclusion

This paragraph feels like a summary, please write a more specific and general conclusion.

Comments on the Quality of English Language

Even if I am not qualified to assess the quality of English in this paper. Minor editing of English language required.

Author Response

Material and methods

Most of suggestions on this topic have been included in the manuscript, including soil analysis, the amount of fertilizer and water applied, as well as the timing and criteria for berry thinning. Regarding bunch thinning, it was not carried out in this experiment.

Results and discussion

All suggestions on this topic have been included in the manuscript.

Why did the Authors at a certain point miss information about 2023.2 production cycle? (Table 4 and in the discussion).

Response: The 2023.2 chemical analysis samples were mixed in the laboratory; for this reason, we chose not to include information about soluble solids (SS), titratable acidity (TA), and the SS/TA ratio.

Conclusion

Comment: "This paragraph feels like a summary, please write a more specific and general conclusion"

Response: We change and improve Conclusion

Quality of English language:

We have already paid for an English editing service for this version of the manuscript

Reviewer 2 Report (New Reviewer)

Comments and Suggestions for Authors

As some of the text in this manuscript is in red font, I assume that this version of the manuscript is a revised version of an earlier submission.

This is a fairly simple study, looking at the performance of eight different rootstocks on grape productivity in the cultivar, BRS Tainá.

Including own-rooted BRS Tainá, (i.e. no rootstock) would have added value to the study. Effectively these would have been control plants. This is an unfortunate omission.

As some readers may not be familiar with this grape growing region, and to better understand why two cropping samples were achieved in one calendar year, indicate if the grape growing region in this study is tropical, sub-tropical etc and explain that under these climatic conditions, two harvest can be obtained per year.

The comments that I have made are minor ones relating to the phrasing of the text. These comments are indicated on the uploaded amended version of the manuscript.

Comments on the Quality of English Language

Overall use of the English language is generally good and the manuscript is presented clearly. I’ve made some minor suggestions in the attached version of the manuscript to improve the wording in places.

Author Response

Introduction

All suggestions on this topic have been included in the manuscript.

 Material and methods

Most of the suggestions on this topic have been made. Only the suggestion below was not included in the manuscript.

Comment: Include details of which fungicides were applied and when.

Response: We did not consider it necessary to include this level of detail in the manuscript, as despite being registered products, the list of fungicides and insecticides used during crop management is very extensive. Additionally, the application frequency is at least once a week, but it can be increased to up to three times a week when weather conditions are favorable to phytopathogens.

This manuscript is a resubmission of an earlier submission. The following is a list of the peer review reports and author responses from that submission.

Round 1

Reviewer 1 Report

Comments and Suggestions for Authors

 Dear Authors,

The manuscript gives interesting information regarding production, number of bunches per plant, berry length and firmness, the physicochemical characteristics of the fruit, by the use the most suitable rootstock for grafting the ‘BRS TAINÁ’ grape variety, in tropical semi-arid conditions.

The article is well structured, and all key elements are present.

The title doesn’t describe very clearly the article.

The abstract content reflects the content of the article.

In the introduction paragraph the authors clearly present the problem investigated but why do the authors think that this use of experimentally selected rootstocks may interfere with the production components of the grafted variety? What’s the theoretical foundation? Authors need to make this information clear in introduction. There should be solid literature to support the research in the introduction.

The purpose of the study is well specified.

There are some issues with the English in places and there are methods that need to be clarified.

Material and methods

I don't think we can call the rootstock used for grafting as a treatment. I think that the rootstock used is the main experimental factor of the research. In the case of this paper, this factor must be the one to which all determinations and discussion are related.

Why did bud load vary between 20 and 30 buds/m2?  This greatly influenced the production of grapes, their number and size per plant, due to the different number of buds left at pruning.

If the vine plants "BRS Tainá" were one year old in the first production cycle, their age most influenced the morphological and productive characteristics in the production cycles of 2022 and 2023, compared to the rootstocks used for grafting!!

If the plot was composed of four plants and only one of the two central plants of each plot was evaluated, I think it is too few to have statistically assured results.

The results are well presented and discussed, but it needs to be emphasized what contributes most to the growth of berries and the quality and quantity of production when different rootstocks are used?

The authors need to discuss the relationship between the dissimilar characteristics of the rootstocks and their findings. Consistency with the hypothesis and objective must be maintained in the item “Results and Discussion”.

Conclusions are supported by the results and are reasonable, but I think the authors should refer to all the rootstocks used, not only to Paulsen 1103.

References are appropriate.

 Thank you!

Comments on the Quality of English Language

I think it should be beter to used:

- the term - factor-  for rootstock instead of treatment

- sometimes the phrase would be clearer with term - means values - instead of means

Author Response

The lines 134 to 136 show the superior characteristics of rootstocks responsible for high yield. The bunch length, bunch width, berry mass, and berry diameter were not influenced when different rootstocks were used, confirming that the climatic variations between the production cycles were more responsible for these changes compared to the main factor (rootstocks). The same results were obtained for the physicochemical variables of ‘BRS Tainá’ berries, in which the overall mean values considering all production cycles did not show significant changes between the rootstocks used.

The English translation was reviewed again but we can provide a new translation by a native English specialist if necessary.

Thanks for all reviewers for the important contributions.

Reviewer 2 Report

Comments and Suggestions for Authors

The topic, rootstock-scion interaction is interesting for a wide audience.

The question is original and well defined: Rootstock-Scion Interaction on Fruit Yield and Quality of ‘BRS 2 TAINÁ’ Seedless Table Grape in Tropical Semi-Arid Conditions.

The cited references are relevant to the research. There are a lot of self-citation, but it is justified by the fact that the group has significant results in the field of Rootstock-Scion Interaction in Tropical Semi-Arid Conditions. The relatively high portion of self-citations, allows the readers to learn the extended research going on this field in Brazil.

The presented work is a part of a series of experiments. The manuscript has an overlapping specially in materials and methods with a previous publication (OLIVEIRA et al., 2023). This previous publication covers the first part of the experiment with “BRS Tainá” variety, where no significant changes were observed in the vigour and fertility of buds from ‘BRS Tainá’ when grafted on different rootstocks. The present manuscript is the second part of the experiment focusing on the effect of different rootstocks on the production components and physicochemical characteristics of ‘BRS Tainá’ grapes. Considering the results, there is no overlapping with the previous work. Results presented in this manuscript are original.

Lane 78-80: Cultural practices related to scion management, as well as prevention and control of pests and diseases, were carried out according to recommendations for regional conditions (SOARES; LEÃO,)   Please add additional reference. 

The cited literature is a 700 pages book, and not the description of the methods used. This citation is not enough to help readers to reconstruct the experiment. As table grape production has a sophisticated technology, please cite a previous work describing the proper technology (possible in English) or add a short description like in the paper of Leão and Oliviera, 2023.

Lane 95-96:  number of bunches per plant (NB), obtained by counting; bunch mass (BM), in g, Indicate, that NB is the number of bunches per plant before thinning, It seems, that the mean bunch weight (g) (BM) was obtained considering the total bunch weight and the number of bunches left (after scion management) per plant.

 Lane 142-144: Table 2 : The number of bunches are too high to get the expected 15kg production per plant. The NB is probably the initial number of bunches before thinning. I suppose, that clusters were thinned to around 50 clusters per vine, and clusters left on the vine were thinned to 300-340 g weight, as suggested earlier (Leão et al. 2021a). The applied technology needs a short description. Indicate that NB is the initial number of bunches, before thinning.

Lane 186: Title of the fig. ”production cycle” is missing.

The paper can not be accepted in the present form.

Author Response

We agreed with the reviewer and considered all of suggestions in the revised version.

The English translation was reviewed again but we can provide a new translation by a native English specialist if necessary.

Thanks for all reviewers for the important contributions.

Reviewer 3 Report

Comments and Suggestions for Authors

In this manuscript authors investigated the agronomic performance of cv. BRS Taina grafted on eight different rootstocks, under the conditions of the Sub-Middle São Francisco Valley, Brasil.

From the scientific point of view, the research is very basic, with only few basic agronomical traits investigated (some yield components and basic grape juice composition), it lacks scientific novelty and it is not suitable for publishing in Plants journal, and it is more adequate for publishing in a local technical journal in the field of agriculture/viticulture.

Moreover, the manuscript is not written according to the guides to authors (style of writing of references), the order of sections, etc.

Although in the title it is mentioned the rootstock-scion interaction, no data about the statistical interaction is presented in the manuscript (and only one scion was used, so it is not possible to asses the interaction). The general style of writing is of quite low standard for a scientific journal.

Comments on the Quality of English Language

The English language should be revised.

Author Response

We agreed with some reviewer’s suggestions and incorporated them into the revised version. The title was changed and the guides to authors were followed.

It is important to note that ‘BRS Tainá’ is a table grape cultivar, which is why we investigated the effect of eight different rootstocks on 12 agronomical traits related to yield components and grape juice composition.

The English translation was reviewed again but we can provide a new translation by a native English specialist if necessary.

Thanks for all reviewers for the important contributions.

On the other hand, the interactions mentioned in the manuscript are season X rootstocks and it is not scion X rootstocks. We think important study the intraction season xrootstocks because the vine response in tropical viticulture is different in the first and seconf harvest of the year.

Round 2

Reviewer 1 Report

Comments and Suggestions for Authors

I agree with all changes made to the paper by the authors and consider that it can be accepted in its present form for publication

Reviewer 3 Report

Comments and Suggestions for Authors

In my opinion the manuscript is still lacking scientific novelty and it is not suitable for publishing in Plants journal. Although the interaction between rootstocks and production cycles is now mentioned at several places in the manuscript, no data on statistical interaction are presented in tables or figures. The general style of writing is still of quite low standard for a scientific journal.